# Microalgae as Functional Ingredients in Savory Food Products: Application to Wheat Crackers

**DOI:** 10.3390/foods8120611

**Published:** 2019-11-23

**Authors:** Ana Paula Batista, Alberto Niccolai, Ivana Bursic, Isabel Sousa, Anabela Raymundo, Liliana Rodolfi, Natascia Biondi, Mario R. Tredici

**Affiliations:** 1LEAF—Linking Landscape, Environment, Agriculture and Food, Instituto Superior de Agronomia, Universidade de Lisboa, Tapada da Ajuda, 1349-017 Lisboa, Portugal; ana.batista@supagro.fr (A.P.B.); isabelsousa@isa.ulisboa.pt (I.S.); anabraymundo@isa.ulisboa.pt (A.R.); 2Department of Agriculture, Food, Environment and Forestry (DAGRI), University of Florence, Piazzale delle Cascine 24, 50144 Florence, Italy; liliana.rodolfi@unifi.it (L.R.); natascia.biondi@unifi.it (N.B.); mario.tredici@unifi.it (M.R.T.); 3Fotosintetica & Microbiologica S.r.l., Via dei Della Robbia 54, 50132 Florence, Italy

**Keywords:** crackers, microalgae, physical properties, antioxidants, phenolics, proteins, digestibility, sensory analysis

## Abstract

Crackers are widely consumed snack foods and there is an increasing trend in adding functional ingredients to their composition. In the present work, the dried biomasses of four microalgae strains—*Arthrospira platensis* F&M-C256, *Chlorella vulgaris* Allma, *Tetraselmis suecica* F&M-M33, and *Phaeodactylum tricornutum* F&M-M40—were used as a source of proteins, antioxidants, and other bioactive molecules in artisanal wheat crackers. Two incorporation levels were tested: 2% (*w/w*) and 6% (*w/w*). The impact of microalgae addition was evaluated in terms of physical properties, biochemical composition, antioxidant activity, in vitro digestibility, and sensory characteristics. Microalgae crackers presented stable color and texture throughout eight weeks of storage. Microalgae crackers were slightly thinner and lighter than the control but presented a similar density in agreement with scanning electron microscope images, indicating that gas retention was not greatly affected by microalgae addition. Regarding biochemical composition, 6% *A. platensis* and *C. vulgaris* crackers presented a significantly higher protein content (13.2–13.5%), for which they could be claimed to be a “source of protein” according to the Regulation (EC) No. 1924/2006. *A. platensis* crackers showed the highest antioxidant activity and attained better sensory analysis scores. *T. suecica* and *P. tricornutum* crackers showed high phenolic content and antioxidant activity but attained low sensory scores mainly because of their unattractive fishy off-flavor.

## 1. Introduction

The world snack food market has been projected to reach USD 639 billion by 2023, growing at a compound annual growth rate (CAGR) of 5.8% from 2018 to 2023 [1]. In recent years, a major market trend has been the introduction of new products containing health-promoting ingredients associated with a growing demand for healthy snacks due to the consumer mind set changing taste and preference. In particular, savory snacks, namely, salted crackers, are growing at a greater rate than sweet snacks [2]. Crackers are considered popular snack products which have appreciable demand amongst consumers [3,4]. Crackers are generally defined as dry, thin, and crisp bakery products [5] usually made of wheat flour, fat (or shortening), salt, and leavening agents (yeast, chemical leaveners, or a combination of both). In general, there are three major types of crackers: saltine (soda crackers), savory, and chemically leavened crackers [6]. While saltine and savory crackers typically involve a “sponge-and-dough” fermentation (~24 h) using a sourdough starter and yeast, chemically leavened crackers do not require this step, and their processing is easier to manage [6]. In addition to leavening agents, enzymes (proteinases) can be incorporated into the formulation of “enzyme crackers” [7]. These snacks are available in different varieties and flavors, which subsequently has fueled their popularity in the global market [8]. Moreover, as baked products, crackers are perceived to be healthy snack option in relation to oil-fried or sugar-filled alternatives [9]. Increased consumer focus on healthy and functional foods has led to the development of low-fat, low-salt, and gluten-free crackers [5]. In response to the growing demand for nutraceutical bakery products, several natural feedstocks, namely, food industry by-products, have also been studied as a source of bioactive molecules (e.g., phenolics, carotenoids, fibers, and minerals) to fortify crackers (e.g., [8,10,11]). 

As documented by numerous studies, microalgae are one type of the potential ingredients that can be used to enhance the nutritional value of food [12,13,14], in particular of cereal-based products such as cookies [15,16,17], bread [18,19], pasta [20,21,22], and extruded snacks [23]. To our knowledge, no study with microalgae-enriched crackers has been published so far.

The high potential of microalgae (including cyanobacteria) as a food ingredient is due to their high nutritional value and potential health benefits [24,25]. Microalgae can have high protein content with a balanced amino acid composition [26] and good content of vitamins, minerals, short- and long-chain polyunsaturated fatty acids, carotenoids, enzymes, and fiber [27,28]. According to a Markets & Markets Report [29], the algae products market, estimated at USD 3.98 billion in 2018, is projected to grow at a CAGR of 5.4% in the period 2018–2023, driven by growing consumer awareness regarding the health benefits of algae-based products and increased preference for vegetarian and vegan foods.

The present work was undertaken to formulate and study the influence of the introduction of microalgae on the physical aspects, biochemical composition, in vitro digestibility, and antioxidant and sensory properties of wheat-based crackers. A considerably higher concentration of microalgae biomass (up to 6% *w/w*) than that found in commercial algal products (generally < 1% *w/w*) was used in order to provide significantly higher levels of bioactive compounds. Four microalgal strains—*Arthrospira platensis* F&M-C256, *Chlorella vulgaris* Allma, *Tetraselmis suecica* F&M-M33, and *Phaeodactylum tricornutum* F&M-M40—were selected as potential food ingredients due to their nutritional quality. *A. platensis* (commonly known as “spirulina”), which has been consumed by human populations since ancient times [30,31], and *C. vulgaris* are widely utilized as nutritional supplements [32] and have already been approved as foods, considering that they have been used to a significant degree for human consumption within the European Union before 15 May 1997 [33]. *A. platensis*, which is particularly rich in proteins, iron, γ-linolenic acid, bioactive sulphated polysaccharides, and phycocyanin [34], represents a promising source for the development of functional foods and beverages (e.g., [35]). Bigagli et al. [36] have shown that *A. platensis* F&M-C256 could be used to prevent and manage dyslipidemias. *C. vulgaris* is also rich in proteins, vitamin B-12, pigments, and glucans, which can boost the immune system [37]. Interest in *Tetraselmis* strains, namely, *Tetraselmis chuii*, which has recently been authorized for commercialization as a food supplement and to prepare *T. chuii*-based sauces and condiments, is also increasing [38]. *T. suecica*, which is used in the present study, has been reported to have a high content of polyunsaturated fatty acids and α-tocopherol [39]. *P. tricornutum* contains a high content of eicosapentaenoic acid (EPA 20:5 ω3), fucoxanthin, and other carotenoids, which are associated with antioxidant, anti-diabetes, and anti-obesity effects [40,41]. Currently, there is an ongoing novel food application of the Swedish company Simris Alg AB for EPA-rich oil from *P. tricornutum* [42], for which extensive toxicological data are required. In a work by Niccolai et al. [43], where the general objective was to evaluate the toxicity of microalgal and cyanobacterial strains on human dermal fibroblasts and *Artemia salina*, the four strains used in this work were tested. *A. platensis* F&M-C256, *T. suecica* F&M-M33, and *C. vulgaris* Allma showed no toxicity, while *P. tricornutum* F&M-M40 exhibited toxicity towards fibroblasts.

## 2. Materials and Methods

### 2.1. Microalgae Strains and Biomass Origin

*Arthrospira platensis* F&M-C256 and *Tetraselmis suecica* F&M-M33 biomasses were provided by Archimede Ricerche S.r.l. (Camporosso, Imperia, Italy) and *Phaeodactylum tricornutum* F&M-M40 was produced at the facility of Fotosintetica & Microbiologica S.r.l. (Sesto Fiorentino, Florence, Italy). *Chlorella vulgaris* (Allma) biomass was obtained from Allma Microalgae (Lisbon, Portugal). *A. platensis* F&M-C256, *T. suecica* F&M-M33, and *P. tricornutum* F&M-M40 were cultivated in GWP^®^-II photobioreactors [44] in semi-batch mode, and then the biomasses were harvested by filtration or centrifugation, frozen, lyophilized, and stored at −20 °C until being analyzed. *A. platensis* F&M-C256 biomass was washed with tap water to remove excess bicarbonate before being frozen. The two marine strains (*T. suecica* F&M-M33 and *P. tricornutum* F&M-M40) were cultivated in F medium [45], while *A. platensis* F&M-C256 was cultivated in Zarrouk medium [46] and *C. vulgaris* Allma was cultivated in a freshwater medium. The biochemical composition of the different biomasses, which were determined as reported in Abiusi et al. [47], is presented in Table 1.

### 2.2. Cracker Preparation

Cracker samples were prepared according to the following control formulation (*w/w*): 60.5% commercial all-purpose wheat flour T55 (76.7% carbohydrates, 7.8% protein, 2.9% fiber, and 1% lipids), 1.5% baking powder (corn starch, sodium diphosphate, and sodium bicarbonate), 1% salt (NaCl), 1% sugar (sucrose), 7.5% vegetable oil, and 28.5% distilled water. Microalgae biomass was added to the same formulation at 2% and at 6% (*w/w*) incorporation levels by replacing a corresponding amount of wheat flour. The ingredients were weighed based on a 150 g batch and kneaded for 1 minute at speed 4 in a food processor (Bimby, Vorwerk, Germany) to obtain a homogeneous cohesive dough. For replications, three separate batches of each cracker formulation were made. The dough was further laminated into thin sheets which were passed through a pasta roller machine under three different gauge positions (the dough was passed through three times for each position). A square mold (56 × 56 mm) was used to cut the laminated dough into pieces, which were then slightly perforated. The shaped cracker dough was left to proof for 10 min at room temperature. The crackers were then baked in a forced-air convection oven (Unox, Italy) at 180 °C for 10 min. Baked samples were dried at 60 °C for 30 min, cooled at ambient temperature for 30 min, and stored in non-transparent closed plastic containers. Physical analyses were performed after 24 h (“week 0”) and after four- and eight-week storage (color, texture, and a_w_). Part of the cracker batches were immediately crushed to powder (using an electric mill) and frozen to be used for biochemical composition, antioxidant capacity, and in vitro digestibility analyses. 

### 2.3. Cracker Analyses

#### 2.3.1. Dimensions

The characteristic dimensions of the crackers were measured using a digital caliper model 684132 (Lee Tools, Houston, TX, USA). The individual width (W) and thickness (T) of 20 crackers from each formulation type were measured; spread ratio (W/T) was calculated accordingly. The weight of the cracker samples was also measured and the corresponding densities calculated (weight (g)/volume (cm^3^)). All these analyses were carried out 24 h after cracker preparation. 

#### 2.3.2. Color Analysis

The color of the cracker samples was measured instrumentally using a Minolta CR-400 (Japan) colorimeter with standard illuminant D65 and a visual angle of 2°. The results were expressed in terms of L*, lightness (values increasing from 0% to 100%); a*, redness to greenness (60 to −60 positive to negative values, respectively); and b*, yellowness to blueness (60 to −60 positive to negative values, respectively) according to the CIELab system. Chroma, C*_ab_ (saturation) and hue angle, hº_ab_, were also calculated as defined by C*_ab_ = [(a*^2^ + b*^2^)]^1/2^; hº_ab_ = arctan(b*/a*). The total color difference between sample crackers throughout the storage time (up to eight weeks) was determined using average L*a*b* values according to ΔE* = [(ΔL*)^2^ + (Δa*)^2^ + (Δb*)^2^]^1/2^. The measurements were conducted under the same light conditions using a white standard (L* = 94.61, a* = −0.53, and b* = 3.62) under artificial fluorescent light at room temperature, and measurements were replicated ten times for each formulation sample (one measurement per cracker) as well as for the control 24 h, 4 weeks, and 8 weeks after preparation.

#### 2.3.3. Texture analysis

Instrumental texture analysis was carried out in a TA.XTplus (Stable Micro Systems, Godalming, UK) texturometer. Crackers were submitted to a “three-point bending” or “snap” test, at 3 mm/s probe speed, with a 5 kg load cell, and at a controlled (20 ± 2 °C) room temperature. Hardness was calculated as the force peak (N) in the force versus time texturogram; N corresponds to the maximum force that is needed to break or “snap” the cracker. Measurements were repeated ten times for each formulation sample (one measurement per cracker) as well as for the control 24 h, 4 weeks, and 8 weeks after preparation.

#### 2.3.4. Microscopy Analysis

Crackers with 2% (*w/w*) microalgae biomass (and the control) were analyzed using a table top scanning electron microscope (TM3000, Hitachi, Tokyo, Japan) under low vacuum (30–50 Pa) conditions and Secondary Electrons/Back-Scattered Electrons (SE/BSE) dual signal mode (combined topography/morphology) with 40×, 100×, and 250× magnification. Cracker samples were cut into small pieces (approximately 2 × 4 × 4 mm) and placed in a sample holding chamber in a cross-sectional position. 

#### 2.3.5. Total Water Content and Water Activity (a_w_) Determination

Crackers samples were analyzed for total water content and water activity. Water content was determined gravimetrically using an automatic moisture analyzer (PMB 202, aeADAM, UK) at 130 °C until a constant weight was achieved. Water activity (a_w_) was determined using a thermohygrometer (HygroPalm HP23-AW, Rotronic AG, Bassersdorf, Switzerland) at 20 ± 1 °C. Measurements were performed during storage time (24 h, 4 weeks, and 8 weeks after preparation) by crushing the samples to powder and using an electric mill immediately before analysis (in triplicate).

#### 2.3.6. Proximate Biochemical Composition Determination

The crackers’ proximate biochemical composition was analyzed on powdered samples in terms of total ash content, crude protein [48], crude fat [49], and total dietary fiber [50] by standard methods, as detailed in Batista et al. [17]. Chemical analyses were performed in triplicate, except for total dietary fiber, where a single test was carried out according to the AOAC 991.43 [50] Standard Method. Carbohydrates were calculated by difference and energy value was calculated using the conversion factors designated in Annex XIV of Regulation (EU) No. 1169/2011 [51]. Sodium content was also analyzed, in triplicate, by atomic absorption spectrometry in a certified external laboratory [52].

#### 2.3.7. Phenolics and Antioxidant Capacity Determination

A total phenolic content assay was carried out according to Ganesan, Kumar, and Bhaskar, [53] using the Folin Ciocalteu assay. Samples of 0.1 g of lyophilized crackers were dissolved in 6 mL deionized water and sonicated (MicrosonTM XL2000, Misonix Inc., Farmingdale, NY, USA) for 30 min at maximum power (frequency 20 kHz and power 130 W) while maintaining the temperature below 30 °C by immersing the sample flask in an ice bath. To 100 µL aliquots of each sample was added 2 mL of 2% sodium carbonate (Sigma-Aldrich, St. Louis, MO, USA) in water. After 2 min, 100 μL of 50% Folin Ciocalteu reagent (Sigma-Aldrich, St. Louis, MO, USA) was added. The reaction mixture was incubated in darkness at 25 °C for 30 min. The absorbance of each sample was measured at 720 nm. Results were expressed in gallic acid equivalents (mg GAE g^−1^) through a calibration curve (gallic acid: 0 to 150 μg mL^−1^, *R*^2^ = 0.9907) (Sigma-Aldrich, St. Louis, MO, USA).

To evaluate the radical scavenging capacity of the cracker samples, a DPPH radical scavenging assay was carried out according to Rajauria et al. [54]. Sample extracts were prepared by adding 0.2 g of lyophilized crackers to 20 mL of a 1:5 methanol:water solution and sonicating for 30 min (the same conditions as described above for total phenols extracts). The DPPH assay was performed by mixing 100 µL of DPPH radical solution (165 µM in methanol, Sigma-Aldrich, St. Louis, MO, USA) and 100 µL of sample extract. The reaction mixtures were incubated in darkness at 30 °C for 30 min and the absorbance was measured at 517 nm. The antioxidant capacity of the samples was expressed in terms of µg of vitamin C equivalent antioxidant capacity (VCEAC) per milligram of sample (ascorbic acid calibration curve: 0 to 10 µg mL^−1^, *R*^2^ = 0.9918) and corresponding radical scavenging activity (RSA). Two blank assays, one without samples and another without reagents, were also performed.

Analyses were repeated in triplicate and performed on powdered cracker samples.

#### 2.3.8. In Vitro Digestibility Tests

The in vitro dry matter digestibility (IVDMD) of the microalgae samples was assessed by the method described in Niccolai et al. [55]. This method reproduces the chemical-enzymatic digestion (by gastric and pancreatic juices) which occurs in the proximal tract of the monogastric digestive system. After a two-step enzymatic digestion with porcine pepsin and pancreatin (AppliChem GmbH, Darmstadt, Germany), the IVDMD (%) was calculated from the difference between the initial sample biomass (1 g) and the undigested biomass residue (after correction for the blank assay). A control assay with casein standard (Sigma-Aldrich, St. Louis, MO, USA), used as reference material with 100% digestibility, was also carried out in order to assess the validity of the method (IVDMD = 97.2 ± 0.8%).

The protein content of the undigested biomass residue was analyzed using Lowry’s [56] method to calculate the in vitro protein digestibility (IVPD, % = total protein content − protein content after digestion) ÷ total protein × 100). 

The same procedure was applied to analyze the IVDM and IVPD of the cracker samples produced in this work and of six commercial cracker samples (Table 2) in order to assess the range of variation of these parameters in commercial products. 

All analyses were repeated in triplicate and performed on powdered samples. 

#### 2.3.9. Sensory Analysis

Cracker samples with 2% microalgal biomass, as well as the control sample, were tested by an untrained sensory analysis panel (*n* = 30, age: 19–38, gender: 8 M, 22 F). The cracker samples were evaluated in terms of color, smell, taste, texture, and global appreciation (six levels from *“very pleasant”* to *“very unpleasant”*), as well as salt perception (five levels from *“very unsalted”* to *“very salty”*). The buying intention was also assessed, from *“would certainly buy”* to *“certainly wouldn’t buy”* (five levels). The assays were conducted in a standardized sensory analysis room according to the standard EN ISO 8589 [57]. 

### 2.4. Statistical Analysis

Statistical analysis of the experimental data was performed using STATISTICA from StatSoft (version 8.0, Dell, Round Rock, TX, USA), through variance analysis (one-way ANOVA), and by the Scheffé test—post hoc comparison at a significance level of 95% (*p* < 0.05). All results have been presented as average ± standard deviation. 

## 3. Results and Discussion 

### 3.1. Cracker Colour 

Microalgae crackers presented unusual and visually attractive colors, as can be observed from the photos shown in Figure 1. The evolution of color parameters throughout storage time (0, 4, and 8 weeks) in terms of lightness (L*), greenness (a*), yellowness (b*), chroma (C*), and hue (hº) is represented in Figure 2.

In terms of lightness, it is evident that the addition of microalgae biomass resulted in much darker crackers in relation to the control (L* 77.7), with 2% microalga samples showing values around 39–49 and 6% microalga samples values around 30–37.

Regarding a*, microalgae samples showed negative values in the green domain. However, when increasing microalgae concentration to 6% the green chromaticity decreased, reaching a* values which were close to zero or even positive in some cases (Figure 2). Similar behavior was observed in terms of yellow chromaticity, with b* and C* values decreasing when algae concentration increased from 2% to 6%. Similar effects have been reported in the literature for microalgae cookies [15,16,17] and other baked products. These results suggest that the reaction kinetics of pigment degradation, namely green chlorophylls, upon high temperature baking might be dependent on the initial pigment concentration. Additionally, the formation of brown-colored chlorophyll degradation products such as pheophorbides and pyropheophorbide can also impact the visual color perception and consequently the positioning of the sample within the L*a*b* tridimensional space. Besides this, moisture loss and volume changes occurring upon baking can also greatly influence the crackers’ color appearance [58]. As will be discussed later, 6% microalgae samples presented lower dimensions (Table 3) and lower water contents (Table 4), which can influence color perception and accelerate surface browning [58]. Similarly, the formation of brown-colored Maillard reaction compounds can also be favored for 6% microalgae samples which have higher protein contents (Table 5).

Nevertheless, hue angle remained relatively constant for each microalgae sample, independently of the concentration, and was very stable throughout the eight-week storage time. This means that the crackers’ color was deeply related to the microalgae’s pigment profile while its intensity or chromaticity might have been affected by the baking procedure. 

*P*. *tricornutum* samples showed the highest b* values (up to 26) and hue angle close to pure yellow (90–96°), which should be related to the presence of fucoxanthin [59]. On the other hand, *C. vulgaris* and *T. suecica* showed the highest a* values (in modulus) and hue angles closer to pure green (116–118°), which is in agreement with the high chlorophyll content of these chlorophytes [60]. 

All cracker samples showed good color stability throughout the eight-week storage period, with variations being barely detected by the human eye and with ΔE* values being lower than 5except for the *A. platensis* crackers, which showed an ΔE* of 9.3 (for 2% alga).

### 3.2. Cracker Dimensions

Characteristic dimensions of the microalgae crackers are presented in Table 3. 

In general, crackers with 2% algal biomass did not show significant differences (*p* > 0.05) in terms of dimensions (width, thickness, and spread ratio) in relation to the control. However, for the higher microalgae incorporation level (6%) it was observed that there was a significant (*p* < 0.05) reduction in the crackers’ width and particularly thickness (3.5–3.8 versus 4.5 mm) when compared to the control. These differences led to higher spread ratios for the microalgae crackers which were statistically significant (*p* < 0.05) for 6% *T. suecica* and 6% *P. tricornutum*. The average weight of the crackers was also significantly lower for the 6% algae samples (3.8–4.7 g) in relation to the control (5.8 g), which might also be related to lower moisture retention upon baking. Nevertheless, the crackers’ density remained statistically (*p* > 0.05) constant upon microalgae addition (Table 3), even if smaller values were reported for the 6% samples. This is a very important quality parameter, since low-weight and less dense crackers are usually desired by consumers [61]. This seems to indicate that gas retention in puffed crisp crackers is not hindered by the presence of microalgae. 

### 3.3. Cracker Texture and Microstructure

Texture is one of the most important quality attributes in crackers since consumers highly appreciate a crisp and crunchy texture [62]. One of the most suitable instrumental analysis tests for assessing the texture of these types of brittle food samples is the “three-point bending” or “snap” test, in which the cracker is placed upon two support beams while a third moves down (parallel) into the sample’s middle point causing the sample to fracture into two pieces [63].

Figure 3 presents the hardness results of the cracker samples as determined by the snap test. 

Due to the extremely brittle nature of these samples, the response to breakage was highly variable, and, consequently, the hardness results had relatively high experimental errors (5–13%).

For all the samples, a significant variation in cracker hardness throughout the eight-week study was not observed (*p* > 0.05), which means that the texture properties remained stable, a relevant quality in terms of sensory acceptance by the consumer.

The crackers with *A. platensis*, *C. vulgaris*, and *T. suecica* did not show significant (*p* > 0.05) hardness differences in relation to the control crackers, although there seemed to be a tendency for hardness to increase with microalgae addition, especially for the 6% samples. These results are also in agreement with previous studies with microalgae cookies in which structural reinforcement has been observed upon microalgae addition (e.g., [15,16,17,64]).

In the case of the *P. tricornutum* crackers the inverse behavior was observed, i.e., there was a tendency for hardness to decrease, which was statistically significant (*p* < 0.05) for the 6% sample. In fact, *P. tricornutum* 6% dough samples were more difficult to manipulate since the dough was very elastic. As a result, these crackers became much thinner and lighter (showing a higher spread ratio) (Table 3) as well as less resistant to breakage.

The samples with 2% microalga incorporation, as well as the control, were also analyzed by SEM. From the microscope images (Figure 4) it seems that the control sample presents a denser and more continuous structure, while crackers with microalgae seem to have a slightly more porous and heterogeneous microstructure. 

It seems reasonable to admit that the addition of microalgae might have influenced gluten matrix formation and/or starch gelatinization mechanisms, although to a limited extent. Batista et al. [65] have reported that *Spirulina* biomass impairs starch gelatinization in model gel systems, namely by increasing the gelatinization temperature, which is probably due to competition for water binding zones during the hydration of starch granules. In fact, during dough mixing, the available free water is partitioned between microalgae and wheat flour hydrophilic sites, which might also impair gluten protein network development. It is possible that the formation of a weaker gluten network might result in the collapse of small gas cells into larger cavities, which might influence gas and water trapping during baking.

### 3.4. Cracker Water Content and Water Activity

Water content and/or water activity (a_w_) are critical quality parameters in low moisture foods which strongly influence their crispiness and sensory acceptance. Above a critical a_w_ value, typically around 0.5, the food materials get softer and stale, losing their crispiness [66,67]. 

Table 4 presents the microalgae crackers’ water content and a_w_ value evolution throughout the storage period (at 24 h, 4 weeks, and 8 weeks), and, as expected, these results seem well correlated.

The control cracker presented an initial a_w_ of 0.128 and adding 2% microalgae biomass caused an increase (*p* < 0.05) of up to 0.15–0.16 (except for *C. vulgaris*), which still corresponds to highly crispy products [67]. On the other hand, higher microalgae biomass incorporations (6%) led to significantly (*p* < 0.05) lower a_w_ values of 0.07–0.08 (except for *T. suecica*). The same behavior was observed in terms of water content, although the differences detected were not statistically significant (*p* > 0.05); samples with 2% microalga had initial moisture content (3.3–4.3%) similar to the control (3.7%), while 6% microalgae crackers had lower initial moisture content (2.1–2.9%). A similar behavior in terms of a_w_ was observed by Batista et al. [17] in short-dough biscuits with the addition of 2% and 6% biomass of the same microalgae. As discussed in the previous sections, it is possible that the addition of high concentrations of microalgae biomass could lead to a weaker gluten network which is unable to efficiently trap gas bubbles and water molecules. Hence, a higher moisture loss for the 6% microalgae crackers could have led to a lower moisture content (Table 4) and lower weight (Table 3) for these samples. There was a tendency for total moisture and a_w_ to increase (*p* < 0.05) with time, which was the same pattern undergone by the control (Table 4). This means that, in general, the addition of microalgae biomass does not seem to have a negative impact on the crackers’ shelf life with respect to a_w_ evolution (and consequent crispness loss). In fact, after 8 weeks’ storage, the majority of the samples showed lower a_w_ than the control, with the lowest values observed for *P. tricornutum* (0.10–0.12). For *T. suecica* 2%, a pronounced increase was observed with time with moisture values increasing from 3.3% up to 7.7% and a_w_ values from 0.15 to 0.52.

### 3.5. Cracker Proximate Biochemical Composition

Table 5 presents the proximate biochemical composition of the crackers prepared with microalgae biomass incorporation. 

Regarding mineral content, no significant differences (*p* > 0.05) were found in terms of total ash content (except for *C. vulgaris* 6%) and sodium contents. Nevertheless, it should be noted that there seems to be a tendency for mineral content to increase for higher microalgae biomass incorporations. In terms of sodium content, the highest value was found in *T. suecica* 6% crackers, which presented 1.1% Na, corresponding to 2.6% NaCl (control: 1.9% NaCl). This is in agreement with the higher sodium content in the biomass of this marine microalga (Table 1). These values are comparable to those of commercial wheat crackers (Table 2, 1.8–2.2 g salt/100 g).

In terms of fat content, there were also no significant (*p* > 0.05) differences between the control and microalgae crackers, with crude fat contents ranging from 11.5% to 13.2% (Table 5), similarly to commercial wheat crackers (Table 2, 6.3–18.0%). 

Microalgae-enriched cracker samples showed significantly higher (*p* < 0.05) protein contents than the control (9.8%) (Table 5). Crackers with 2% microalgae ranged from 10.4 to 11.4% protein while 6% microalgae crackers ranged from 12.1 to 14.6% protein, with the highest values obtained for *A. platensis* and *C. vulgaris* samples. This is in agreement with the high protein content of these microalgae (Table 1) and to previous studies in the literature for microalgae-enriched bakery food products (e.g., [17,20,21,23,64]). In fact, these microalgae-enriched crackers can be regarded as a very interesting protein-fortified bakery product. For 6% *A. platensis* and 6% *C. vulgaris* crackers, the protein content was found to correspond to 13.2 and 13.5%, respectively, of the total energy content of the sample, so these products can be claimed as *“source of protein”* according to Regulation (EC) 1924/2006 [68].

Regarding the total dietary fiber content, microalgae cracker samples presented average values ranging from 4.4 to 6.7% in the same order of magnitude as the control crackers (5%) (commercial wheat crackers: 2.5–5.0%, Table 2). The highest value was reported for the *A. platensis* 6% sample. Previous studies have also reported higher fiber content in *A. platensis*–enriched biscuits [64,69]. The total carbohydrate value was always lower than the control, while the energy values remained relatively constant (430–442 kcal/100 g) and similar to those of commercial wheat crackers (Table 2, 410–461 kcal/100 g).

### 3.6. Cracker Phenolic Content and Antioxidant Capacity

Figure 5 represents the total phenolic content of the cracker samples. The addition of 2% microalgae biomass did not promote significant (*p* > 0.05) differences in phenolic content in relation to the control (1.4 mg GAE g^−1^). However, when increasing microalgae biomass content to 6%, phenolic content significantly (*p* < 0.05) increased to 1.6–1.7 mg GAE g^−1^ for *A. platensis* and *C. vulgaris* and to 2.3–2.5 mg GAE g^−1^ for *T. suecica* and *P. tricornutum*, which corresponds to a net total phenolics increase of about 0.3 and 1 mg GAE g^−1^ compared to the control, respectively.

Other authors have highlighted the high phenolic compound content of *Phaeodactylum* and *Tetraselmis* [70,71]. 

The results obtained for *T. suecica* crackers are higher than those attained by Batista et al. [17] in sweet cookies prepared with the same microalgae strain. Similar behavior has been reported by Abraços [72] in *Salvia sclaeroides*-enriched sweet cookies and crackers (ten times higher phenolic content in the crackers) which were prepared exactly under the same thermal processing conditions as the microalgae crackers and biscuits studied in this work.

The antioxidant capacity of microalgae-enriched crackers was tested by the DPPH method (Figure 6). 

Similarly to the total phenolic results (Figure 5) for the 2% microalgae crackers, there were no significant (*p* > 0.05) improvements in the antioxidant capacity (0.55–0.60 µg VCEAC mg^−1^) in relation to the control sample (0.54 µg VCEAC mg^−1^, 53% RSA). The addition of 6% microalgae led to a significant (*p* < 0.05) antioxidant capacity increase (up to 0.62–0.65 µg VCEAC mg^−1^) for the *T. suecica* and *P. tricornutum* samples. Moreover, 6% *A. platensis* crackers showed the highest antioxidant capacity value of 0.70 µg VCEAC mg^−1^, which corresponds to 68% RSA. Other authors [17,64,73] have also reported high antioxidant activity for *A. platensis*-enriched biscuits, which might be related to the presence of phycocyanin. 

### 3.7. In Vitro Digestibility

Figure 7 presents the in vitro digestibility of microalgae crackers, in terms of total dry matter (IVDMD) (Figure 7a) and protein (IVPD) (Figure 7b) digestibility. The lower and upper limits for these parameters, determined for six commercial crackers, are also represented in Figure 7.

Microalgae crackers showed IVDMD values ranging from 82% to 86%, values which were slightly lower than that of the control sample (87.7%). This reduction was only statistically different (*p* < 0.05) for the 6% *C. vulgaris* and 6% *P. tricornutum* samples. It is worth nothing that all the values fall within the range of values for the commercial crackers analyzed (78–93%).

Regarding IVPD, there was a wider dispersion of results, with commercial crackers’ values ranging from 26% to 78% and the control sample having 75% IVPD. Crackers with 2% microalgae biomass showed IVPD values ranging from 50 to 66%, i.e., a bit lower than that of the control but still within the commercial samples range. However, the samples with 6% microalgal biomass showed higher IVPD values, particularly *A. platensis* (83%). On the other hand, 6% *C. vulgaris* sample showed the lowest IVPD (42%); the value was still within the range of that of the commercial samples, however. By comparing IVPD values with the initial protein content of the samples (Table 5), it can be observed that, for example, the 6% *A. platensis* sample, which has 14.3% protein content and 83% IVPD, yields 11.9 g digestible protein per 100 g cracker, which is considerably higher than that of the control (7.3 g digestible protein per 100 g cracker). This high protein availability reinforces the interest of this sample as a “protein source” food. 

Several authors have previously reported that *A. platensis* biomass shows 10% to 20% higher IVDMD [17,74] and up to 50% higher IVPD [32] than *C. vulgaris* biomass. The different IVDMD and IVPD values between *A. platensis* and *C. vulgaris* biomass are probably related to differences in cell wall structure, considering that while cyanobacteria have thin cell walls composed by peptidoglycan, green algae such as *C. vulgaris* have thick and rigid cellulosic cell walls [75]. 

### 3.8. Sensory Evaluation

Sensory analysis trials were carried out on samples with 2% microalgae biomass. Figure 8 represents the average scores of the sensorial parameters as evaluated by the panel. 

The control sample showed high sensory scores (>4) and was preferred over the microalgae crackers, which was as expected for this type of highly innovative product. The texture of the microalgae crackers presented good sensorial scores which were similar to the control, which can be related to the instrumental texture, density, and a_w_/water content results, indicating a crisp and aerated texture. The *A. platensis* sample was preferred in terms of taste, smell, and global appreciation, while crackers with marine microalgae (*T. suecica* and *P. tricornutum*) showed the lowest scores for these parameters, as well as for color. 

Interestingly, the addition of microalgae seems to affect the perception of saltiness, with a higher frequency of “unsalted” classification ascribed to algae crackers (20–33%) in relation to the control (10%) (Figure 9).

Regarding intent to buy (Figure 10), 40% of the panelists declared that they would certainly or probably buy the *A. platensis* samples. 

The lower sensory scores attributed to *T. suecica* and *P. tricornutum* crackers was agreed upon by more than 60% of the panelists, who declared that they would probably or certainly not buy this product. Batista et al. [17] have also reported a preference for *A. platensis*-enriched cookies in relation to *C. vulgaris*. Fradique et al. [76] have also reported lower sensory scores, including “strange flavor” detection for pasta enriched with the marine microalgae *Isochrysis galbana* and *Diacronema vlkianum*.

## 4. Conclusions

In this work, microalgae biomass was successfully used as an ingredient in artisanal wheat crackers. The microalgae crackers presented a stable and attractive crispy and aerated texture that was similar to the control and appreciated by sensory analysis panelists. Crackers with 6% *A. platensis* biomass seem particularly interesting due to their high protein content (making them eligible for a *“source of protein”* claim) and high antioxidant activity, while the 2% formulation presented high digestibility and attained high sensory analysis scores. The use of marine microalgae strains such as *T. suecica* and *P. tricornutum* is interesting due to the high phenolic content and antioxidant activity of the crackers, but low sensory scores and novel food regulation constraints postpone their utilization.

## Figures and Tables

**Figure 1 foods-08-00611-f001:**
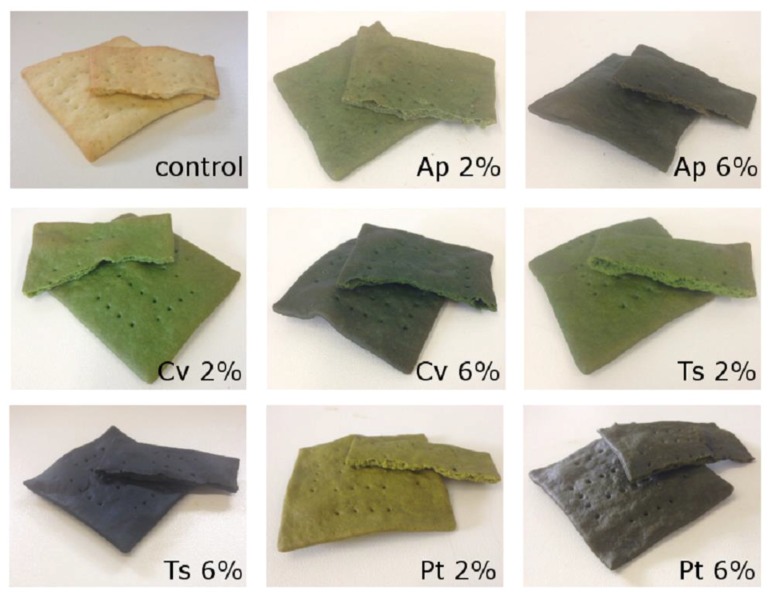
Control cracker and crackers with 2% (*w/w*) and 6% (*w/w*) microalgae biomass. Legend: Ap, *A. platensis*; Cv, *C. vulgaris*; Ts, *T. suecica*; Pt, *P. tricornutum*.

**Figure 2 foods-08-00611-f002:**
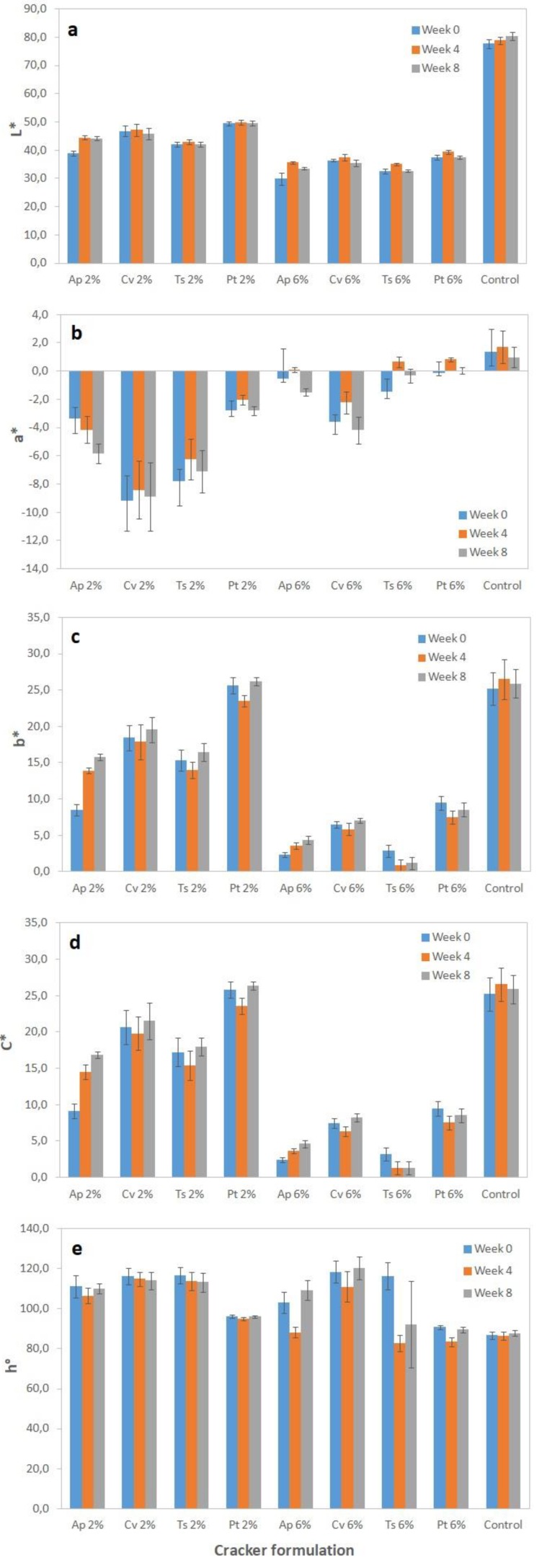
Color parameters lightness (L*) (**a**), yellowness (a*) (**b**), greenness (b*) (**c**), chroma (C*) (**d**), and hue (hº) (**e**) of crackers with 2% and 6% (*w/w*) microalgae biomass incorporation at weeks 0, 4, and 8 Results are expressed as average ± standard deviation (*n* = 10).

**Figure 3 foods-08-00611-f003:**
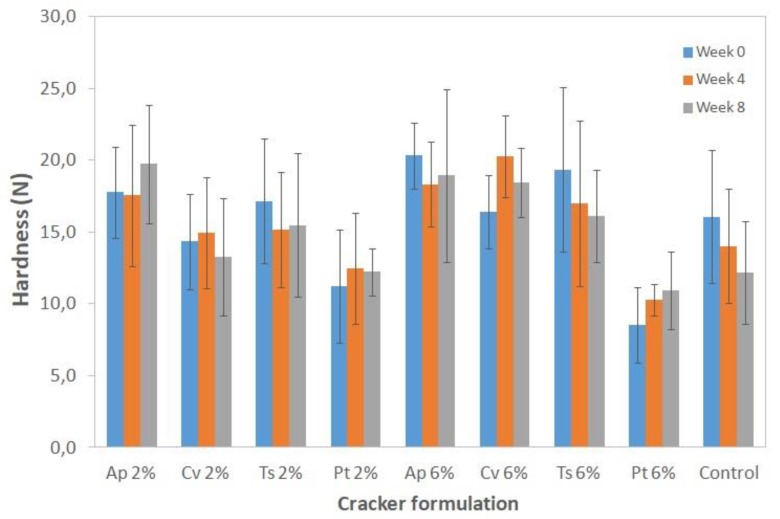
Hardness of crackers with 2% and 6% (*w/w*) microalgae biomass incorporation at weeks 0, 4, and 8 Results are expressed as average ± standard deviation (*n* = 10).

**Figure 4 foods-08-00611-f004:**
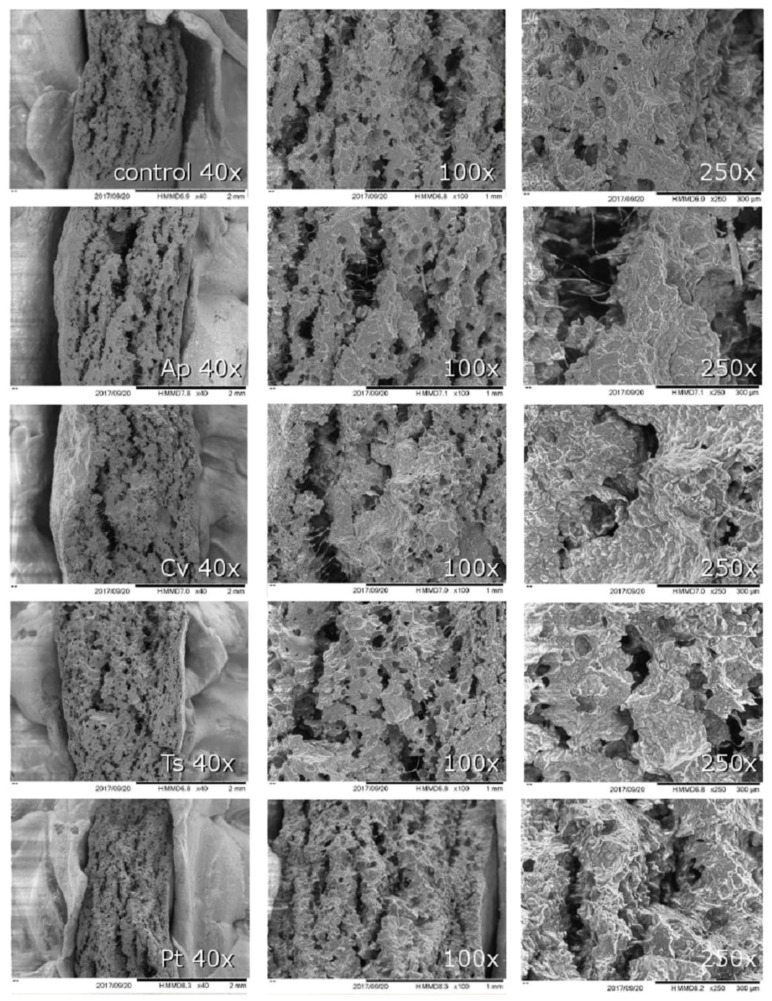
SEM images of crackers with 2% (*w/w*) microalgae biomass incorporation.

**Figure 5 foods-08-00611-f005:**
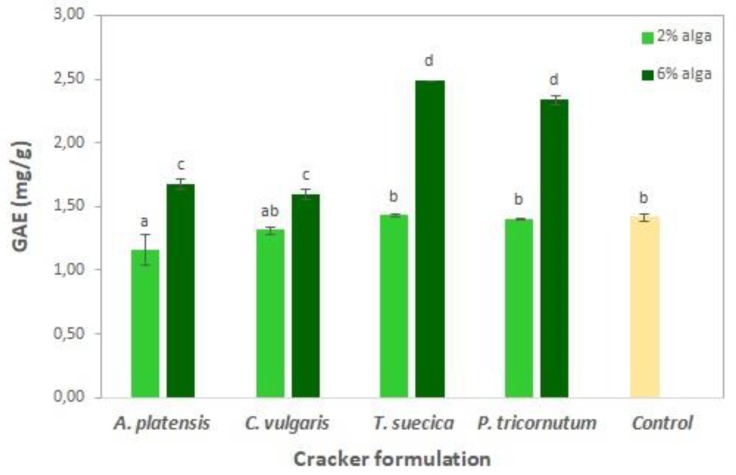
Total phenolic content (expressed as gallic acid equivalents mg g^−1^ dry weight) of crackers enriched with different levels of microalgae incorporation. Results are expressed as average ± standard deviation (*n* = 3).

**Figure 6 foods-08-00611-f006:**
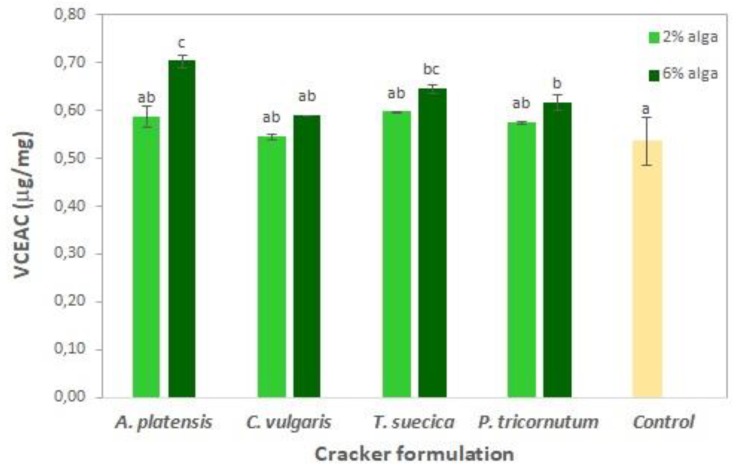
Antioxidant capacity (expressed as µg of vitamin C equivalent antioxidant capacity (VCEAC) per g) of crackers enriched with different levels of microalgae. Results are expressed as average ± standard deviation (*n* = 3).

**Figure 7 foods-08-00611-f007:**
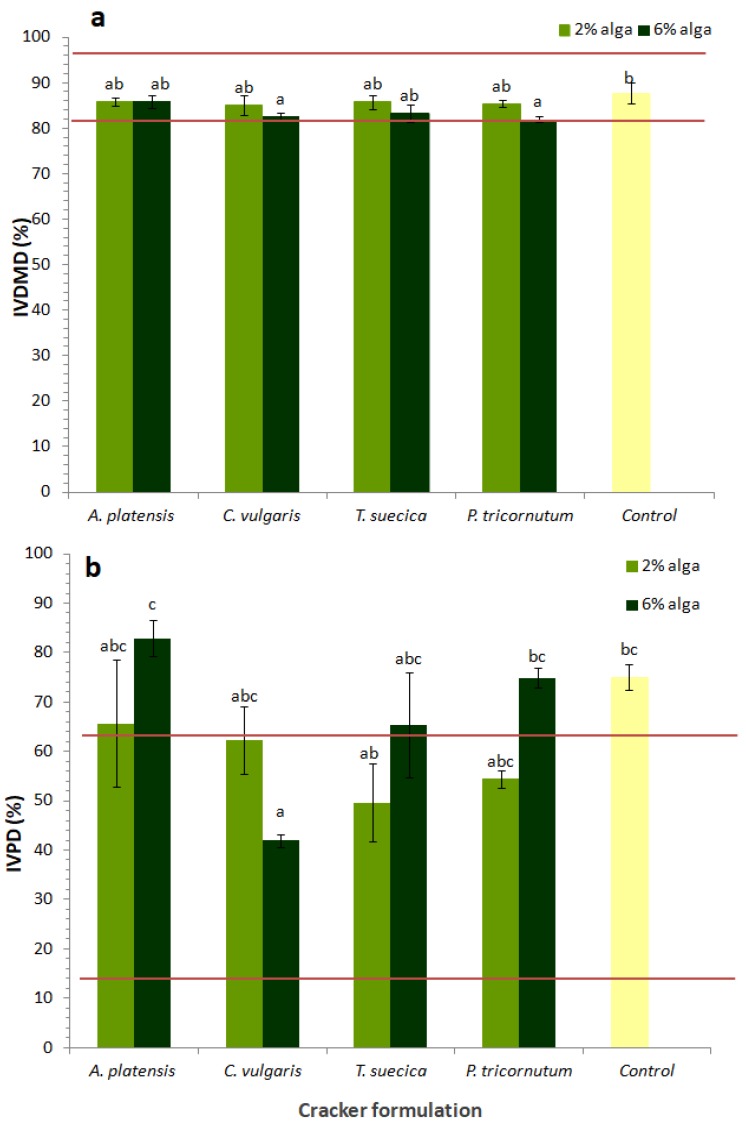
In vitro digestibility (**a**) and in vitro protein digestibility (**b**) of crackers enriched with different levels of microalgae. Results are expressed as average ± standard deviation (*n* = 3). Red lines correspond to the range of variation of six commercial crackers samples tested under the same conditions.

**Figure 8 foods-08-00611-f008:**
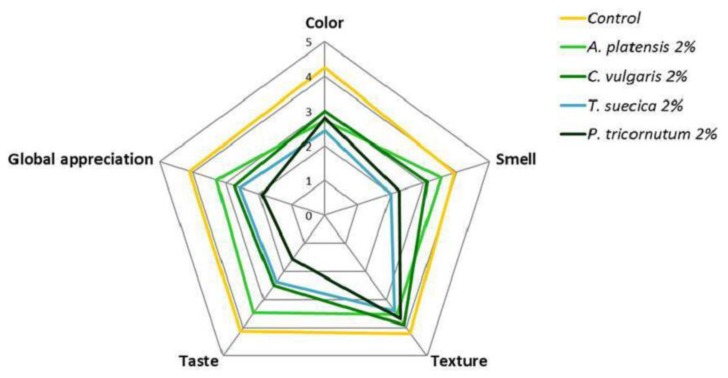
Responses of the sensory analysis panel tasters (*n* = 30) regarding crackers enriched with 2% microalgal biomass, as well as the control sample. Sensory attributes were classified as: 0—very unpleasant; 1—unpleasant; 2—slightly unpleasant; 3—slightly pleasant; 4—pleasant; and 5—very pleasant.

**Figure 9 foods-08-00611-f009:**
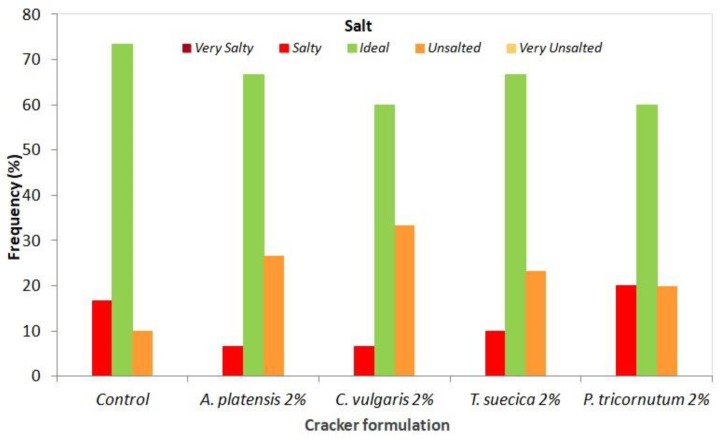
Responses of the sensory analysis panel tasters (*n* = 30) in terms of salt perception of crackers enriched with 2% microalgal biomass, as well as the control sample.

**Figure 10 foods-08-00611-f010:**
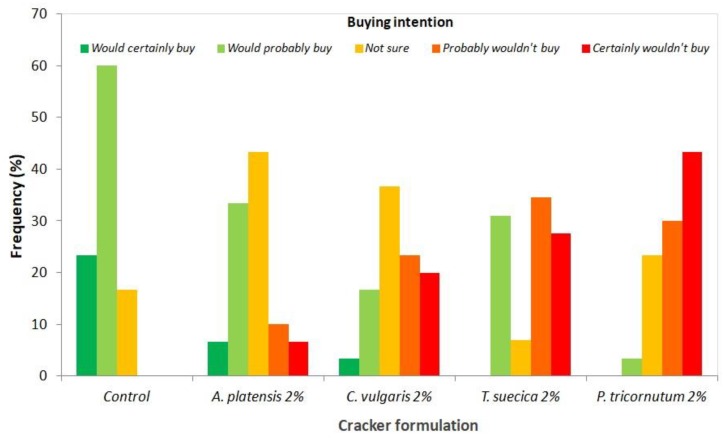
Responses of the sensory analysis panel tasters (*n* = 30) in terms of buying intention for crackers enriched with 2% microalgal biomass, as well as the control sample.

**Table 1 foods-08-00611-t001:** Biochemical composition of the four microalgae biomasses used in the experiments (%, dry weight). Results are expressed as average ± standard deviation (*n* = 3).

Biochemical Component	*A. platensis* F&M-C256	*C. vulgaris* Allma	*T. suecica* F&M-M33	*P. tricornutum* F&M-M40
Protein (%)	68.9 ± 1.0	56.8 ± 2.7	40.2 ± 0.5	38.8 ± 0.1
Lipid (%)	10.7 ± 0. 6	16.9 ± 2.8	28.5 ± 1.2	19.3 ± 1.7
Carbohydrate (%)	12.8 ± 0.2	5.9 ± 0.3	10.2 ± 0.2	11.0 ± 0.7
Ash (%)	6.1 ± 0.1	9.3 ± 1.5	15.7 ± 0.2	14.8 ± 0.1
Sodium (%)	0.6 ± 0.1	0.3 ± 0.2	2.1 ± 0.1	1.4 ± 0.2

**Table 2 foods-08-00611-t002:** Description, ingredients, and average nutritional values of commercial cracker samples analyzed for in vitro dry matter digestibility (IVDMD) (%) and in vitro protein digestibility (IVPD) (%).

Product Designation	Brand G	Brand C	Brand I	Brand W	Brand T	Brand F
Product Description	Crackers with olive oil (6.6%) and sea salt	Crackers with extra virgin olive oil (5.4%)	*Cracker italiani bio artigianali*	*Fette croccanti* (crunchy slices)	Crackers with wheat flour and caramelized cereals on the surface (rice, wheat, and barley)	Rice crackers
Ingredients	Wheat flour, potato flour, olive oil (6.6%), sunflower oil, skimmed milk powder, potato starch, modified corn starch, glucose syrup, milk proteins, sea salt (2%), salt, and raising agent (ammonium hydrogen carbonate)	Wheat flour, extra virgin olive oil (5.4%), salt, brewer’s yeast, malt extract (wheat and barley), barley flour, malted barley flour, and acidity corrector (sodium acid carbonate)	Durum wheat semolina flour, extra virgin olive oil (13.5%), organic tomato extract, organic chili pepper extract, organic garlic extract, organic origan essential oil, and organic basil essential oil	Wholemeal rye flour (111g/100g), salt	Wheat flour (80.4%), palm fat, sugar, leveaning agents (amonium bicarbonate, calcium phosphates, and sodium bicarbonate), puffed rice (1.73%), salt, puffed wheat (0.23%), puffed barley (0.74%), crushed rice grains (0.51%), crushed integral wheat grains (0.23%), rice flour (0.04%), wheat gluten, wheat germen, powdered skimmed milk, wheat starch, and barley malt extract	Rice flour (91%), sugar, sunflower oil, antioxidant E320 (BHA), salt, powdered soy sauce (without gluten) (hydrolyzed grain), flavor enhancer E635 (disodium ribonucleotides), palm oil
**Average nutritional values (100 g)**						
Energy	1822 kJ	1732 kJ	1940 kJ	1405 kJ	1850 kJ	1707 kJ
	433 kcal	410 kcal	461 kcal	334 kcal	440 kcal	408 kcal
Fat	13.0 g	6.3 g	14.8 g	1.5 g	18.0 g	3.8 g
(of which: saturated)	2.3 g	0.9 g	1.9 g	0.3 g	8.4 g	1.3 g
Carbohydrate	67.5 g	76.0 g	69.0 g	60.0 g	60.0 g	85.0 g
(of which: sugars)	11.0 g	2.0 g	1.4 g	2.0 g	6.9 g	4.9 g
Fiber	5.0 g	2.5 g	3.6 g	22.0 g	3.3 g	-
Protein	9.0 g	11.0 g	12.4 g	9.0 g	8.6 g	7.1 g
Salt	2.2 g	1.8 g	1.9 g	1.0 g	2.2 g	1.2 g

**Table 3 foods-08-00611-t003:** Characteristic dimensions of crackers with 2% and 6% (*w/w*) microalgae biomass incorporation. Results are expressed as average ± standard deviation (*n* = 20). Different letters in the same column correspond to significant differences (*p* < 0.05).

Cracker Formulation		Width (W) (mm)	Thickness (T) (mm)	Spread Ratio (W/T)	Weight (g)	Density (g/cm^3^)
Control		52.9 ± 0.6 ^de^	4.5 ± 0.5 ^c^	11.9 ± 1.4 ^a^	5.8 ± 0.6 ^e^	0.48 ± 0.06 ^ab^
*A. platensis*	2%	52.5 ± 0.5 ^cde^	3.7 ± 0.2 ^ab^	14.2 ± 1.1 ^abc^	5.3 ± 0.3 ^de^	0.51 ± 0.05 ^a^
	6%	51.9 ± 0.6 ^abc^	3.8 ± 0.4 ^ab^	13.7 ± 1.6 ^abc^	4.7 ± 0.3 ^bcd^	0.47 ± 0.05 ^ab^
*C. vulgaris*	2%	52.7 ± 0.7 ^de^	4.0 ± 0.4 ^bc^	13.4 ± 1.3 ^ab^	5.2 ± 0.5 ^de^	0.48 ± 0.05 ^ab^
	6%	51.3 ± 0.7 ^a^	3.7 ± 0.3 ^ab^	13.7 ± 1.1 ^abc^	4.4 ± 0.4 ^ab^	0.45 ± 0.04 ^ab^
*T. suecica*	2%	52.5 ± 0.8 ^de^	3.8 ± 0.5 ^bc^	13.7 ± 1.7 ^ab^	5.0 ± 1.0 ^cde^	0.46 ± 0.05 ^ab^
	6%	51.8 ± 1.1 ^bcd^	3.6 ± 0.6 ^ab^	14.4 ± 2.0 ^bc^	4.3 ± 0.9 ^abc^	0.45 ± 0.03 ^b^
*P. tricornutum*	2%	53.3 ± 0.5 ^e^	4.2 ± 0.5 ^bc^	12.8 ± 1.6 ^ab^	5.5 ± 0.7 ^e^	0.47 ± 0.04 ^ab^
	6%	51.5 ± 0.9 ^ab^	3.5 ± 0.5 ^a^	15.0 ± 2.3 ^c^	3.8 ± 0.5 ^a^	0.43 ± 0.05 ^b^

**Table 4 foods-08-00611-t004:** Evolution of total water content and water activity (a_w_) of crackers with 2% and 6% (*w/w*) microalgae biomass incorporation across time. Results are expressed as average ± standard deviation (*n* = 3). Different letters (*a* to *h*) in the same column correspond to significant differences (*p* < 0.05) between samples. Samples marked with *x*, *y*, or *z* in the same line correspond to significant (*p* < 0.05) differences from week 0 to week 8.

		Water Content (%, *w/w*)			Water Activity (a_w_)		
		Week 0	Week 4	Week 8	Week 0	Week 4	Week 8
Control		3.7 ± 0.1 ^bc,x^	4.7 ± 0.2 ^c,y^	5.4 ± 0.3 ^cd,z^	0.128 ± 0.002 ^b,x^	0.206 ± 0.007 ^c,y^	0.362 ± 0.005 ^a,z^
*A. platensis*	2%	4.1 ± 0.1 ^bc,x^	5.1 ± 0.1 ^c,y^	6.0 ± 0.2 ^d,z^	0.155 ± 0.005 ^cd,x^	0.282 ± 0.001 ^d,y^	0.385 ± 0.004 ^b,z^
	6%	2.9 ± 0.8 ^abc,x^	3.7 ± 0.1 ^b,xy^	4.5 ± 0.1 ^bc,y^	0.081 ± 0.002 ^a,x^	0.176 ± 0.005 ^b,y^	0.270 ± 0.005 ^c,z^
*C. vulgaris*	2%	3.4 ± 0.5 ^abc,x^	4.8 ± 0.1 ^c,y^	4.5 ± 0.1 ^bc,y^	0.126 ± 0.001 ^b,x^	0.207 ± 0.013 ^c,y^	0.289 ± 0.005 ^d,z^
	6%	2.8 ± 0.2 ^ab,x^	4.9 ± 0.2 ^c,y^	5.9 ± 0.2 ^d,z^	0.081 ± 0.003 ^a,x^	0.259 ± 0.010 ^d,y^	0.385 ± 0.003 ^b,z^
*T. suecica*	2%	3.3 ± 0.6 ^abc,x^	6.2 ± 0.2 ^d,y^	7.7 ± 0.7 ^e,y^	0.145 ± 0.003 ^c,x^	0.385 ± 0.004 ^e,y^	0.516 ± 0.005 ^e,z^
	6%	2.9 ± 0.1 ^abc,x^	3.4 ± 0.2 ^b,x^	3.3 ± 0.3 ^ab,x^	0.121 ± 0.007 ^b,x^	0.162 ± 0.004 ^b,y^	0.171 ± 0.002 ^f,y^
*P. tricornutum*	2%	4.3 ± 0.1 ^c,x^	3.6 ± 0.1 ^b,y^	3.3 ± 0.2 ^ab,y^	0.159 ± 0.005 ^d,x^	0.127 ± 0.008 ^a,y^	0.120 ± 0.005 ^g,y^
	6%	2.1 ± 0.2 ^a,x^	2.6 ± 0.2 ^a,x^	2.5 ± 0.1 ^a,x^	0.069 ± 0.003 ^a,x^	0.106 ± 0.002 ^a,y^	0.100 ± 0.008 ^h,y^

**Table 5 foods-08-00611-t005:** Proximate biochemical composition (g/100g) of crackers with 2% and 6% (*w/w*) microalgae biomass incorporation. Results are expressed as average ± standard deviation (*n* = 3) except for total dietary fiber (*n* = 1). Different letters in the same column correspond to significant differences (*p* < 0.05).

		Total Ash (g/100g dw)	Sodium (g/100g dw)	Crude Fat (g/100g dw)	Crude Protein (g/100g dw)	Total Dietary Fiber (g/100g dw)	Carbohydrates * (g/100g dw)	Energy Value (kcal/100g dw)
Control		3.3 ± 0.5 ^a^	0.78 ± 0.07 ^a^	12.1 ± 0.8 ^a^	9.8 ± 0.4 ^a^	5.0	69.8	437
*A. platensis*	2%	3.4 ± 0.5 ^a^	0.73 ± 0.17 ^a^	12.5 ± 1.1 ^a^	11.0 ± 0.1 ^bc^	5.6	67.5	438
	6%	4.6 ± 0.5 ^ab^	0.86 ± 0.12 ^a^	12.9 ± 0.8 ^a^	14.3 ± 0.1 ^e^	6.7	61.5	433
*C. vulgaris*	2%	4.4 ± 0.3 ^ab^	0.80 ± 0.07 ^a^	11.5 ± 1.4 ^a^	11.4 ± 0.3 ^c^	5.1	67.6	430
	6%	4.8 ± 1.0 ^b^	0.82 ± 0.04 ^a^	12.3 ± 1.4 ^a^	14.6 ± 0.1 ^e^	5.7	62.6	431
*T. suecica*	2%	3.4 ± 0.1 ^ab^	0.78 ± 0.06 ^a^	13.1 ± 0.1 ^a^	10.4 ± 0.1 ^ab^	4.7	68.5	442
	6%	4.6 ± 0.1 ^ab^	1.06 ± 0.09 ^a^	12.6 ± 0.4 ^a^	12.8 ± 0.4 ^d^	4.4	65.6	436
*P. tricornutum*	2%	3.3 ± 0.1 ^a^	0.81 ± 0.11 ^a^	13.2 ± 0.2 ^a^	10.7 ± 0.1 ^bc^	5.8	67.1	441
	6%	4.3 ± 0.1 ^ab^	0.89 ± 0.08 ^a^	12.7 ± 0.4 ^a^	12.1 ± 0.1 ^d^	6.3	64.6	434

* Carbohydrates were calculated by difference from average ash, fat, protein, and fiber contents.

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
