# Peer review of "Microalgae as Functional Ingredients in Savory Food Products: Application to Wheat Crackers"

_foods, 2019, doi:10.3390/foods8120611_

Round 1

Reviewer 1 Report

Authors presented an excellent study in which demonstrated that several fresh and sea-water microalgae could be used successfully used as ingredient in artisanal wheat crackers on the basis of the physical aspect, biochemical composition, in vitro digestibility, antioxidant and sensory properties of additivated crackers relative to control. Interestingly, a considerably higher concentration of microalgae biomass (2 % w/w) than those found in commercial algal products (generally < 1% w/w) was feasible, improving bioactive properties of crackers relative to control, without significantly diminishing the sensory properties above mentioned relative to control. Despite low sensory scores mainly because their unattractive fishy off-flavour is a potential obstacle to its commercialization, additivated crackers were positively valued by sensory analysis panellists. It is worth highlighting 40% of the panellists declared that they would certainly or probably buy the A. platensis samples. The study is very interesting to be published in the Foods journal. The article is clearly written, well organized and the data are very well presented.

Reviewer 2 Report

line 114  Is this 2% and 6% for the overall formula?  Please clarify.  Also state how many separate batches of each cracker formulation were made for replication.

line 250  Your explanation is not sufficient.  samples were baked the same way so why the decrease in green? Also, what do you mean by pigment saturation?  Please clarify. 

line 343 to 344  These are conflicting concerning moisture levels.

line 357 Table 5  Can you add standard deviations for fiber and carbs and energy to the table?

line 366  What do you mean by corresponding to 2.6% NaCl?  How did you get that number?

line 394  Change to 'differences in total phenolic content in relation to'.  Also refer to figure in text.

line 465  Figure 9  Why do you only show three columns for each sample, but have five levels in the legend?
